# Iron Replacement Therapy with Oral Ferric Maltol: Review of the Evidence and Expert Opinion

**DOI:** 10.3390/jcm10194448

**Published:** 2021-09-28

**Authors:** Carsten Schmidt, Stephen Allen, Nelson Kopyt, Pablo Pergola

**Affiliations:** 1Medical Clinic II, Department of Gastroenterology, Hepatology, Endocrinology, Diabetology and Infectious Diseases, Klinikum Fulda, Pacelliallee 4, 36043 Fulda, Germany; 2Medical Faculty, Friedrich Schiller University, 07747 Jena, Germany; 3Department of Clinical Sciences, Liverpool School of Tropical Medicine, Pembroke Place, Liverpool L3 5QA, UK; stephen.allen@lstmed.ac.uk; 4Department of Medicine, Division of Nephrology, Lehigh Valley Hospital, 1230 S Cedar Crest Boulevard, Suite 301, Allentown, PA 18103, USA; nkopyt@gmail.com; 5Renal Associates PA, 1123 N Main Av., Suite 120, San Antonio, TX 78212, USA; ppergola@raparesearch.com

**Keywords:** adult, anemia, chronic kidney disease, ferric maltol, hemoglobin, inflammatory bowel disease, iron deficiency, pediatric, pulmonary hypertension, tolerability

## Abstract

Iron deficiency is the most common cause of anemia globally and is frequently reported in patients with underlying inflammatory conditions, such as inflammatory bowel disease (IBD) and chronic kidney disease (CKD). Ferric maltol is a new oral iron replacement therapy designed to optimize iron absorption while reducing the gastrointestinal adverse events associated with unabsorbed free iron. Ferric maltol has been studied in clinical trials involving almost 750 adults and adolescents with iron-deficiency anemia associated with IBD, CKD, and other underlying conditions, and it has been widely used in clinical practice. It is approved for the treatment of adults with iron deficiency with or without anemia, independent of the underlying condition, and is commercially available in Europe and the United States. We review the published evidence for ferric maltol, which demonstrates consistent and clinically meaningful improvements in hemoglobin and measures of iron availability (ferritin and transferrin saturation) and shows that it is well-tolerated over long-term treatment for up to 64 weeks—an important consideration in patients with chronic underlying conditions such as IBD and CKD. We believe that ferric maltol is an effective, convenient, and well-tolerated treatment option for iron deficiency and iron-deficiency anemia, especially when long-term management of chronic iron deficiency is required. Writing support was provided by Shield Therapeutics (Gateshead, UK).

## 1. Introduction

Iron deficiency refers to any situation in which there is insufficient iron available in the body to meet physiologic needs. It is the most common nutritional disorder in humans, accounting for around 50% of the estimated 2.2 billion cases of anemia (low hemoglobin) worldwide [1,2,3,4]. Absolute iron deficiency can develop as a result of malnutrition, malabsorption, or blood loss. Inflammatory bowel disease (IBD) is further complicated by absolute iron deficiency due to chronic gastrointestinal blood loss and impaired iron absorption across the damaged bowel mucosa, as well as inflammation-associated (hepcidin-mediated) downregulation of iron absorption [5]. In addition, inflammatory disorders such as IBD and chronic kidney disease (CKD) may result in functional iron deficiency, with impaired iron absorption and release from storage proteins resulting in inappropriate iron availability to meet metabolic demands, such as erythropoiesis [4,6].

Iron deficiency adversely affects overall health and well-being, even before the development of anemia [7,8,9,10,11,12]. Iron deficiency, with or without anemia, is associated with fatigue, impaired physical and cognitive function, headache, tachycardia, and dyspnea [7,8,9,10,11,12,13,14,15,16,17], with resulting adverse impacts on daily activities, productivity, quality of life, and any additional underlying illnesses [7,8,9,10,11,12,13,14,15,18]. The wide-ranging effects of iron deficiency reflect the crucial role that iron plays in oxygen transport, enzymatic reactions, cellular processes, immunity, and cognitive function [4,19,20].

Iron absorption, transport, storage, and use are normally tightly regulated, with daily small losses in the urine, feces, or sweat replaced by iron in the diet [6]. Once absorbed, most iron is incorporated into hemoglobin in erythrocytes, while some is converted to myoglobin in muscle tissue. Iron that is not immediately needed can be stored in ferritin or hemosiderin [21,22,23]. The transport of iron from cell to cell is achieved via iron transporter mechanisms, including ferroportin, which mediates iron transport out of the cell from the cytoplasm across the cell membrane; transferrin, which binds iron in the plasma for delivery to tissues; and transferrin receptors on cell surfaces, which allow the iron to be internalized [6,21,23]. Ferroportin activity is regulated by hepcidin in response to the body’s iron requirements: in the presence of high iron levels, hepcidin is upregulated, binding to ferroportin to stimulate degradation of the transport protein, thereby preventing the movement of iron out of the cell and reducing the amount of available iron [6,22,23]. Hepcidin production is also stimulated by cytokines, resulting in the low availability of iron in inflammatory conditions [21,22,23]. In patients with kidney dysfunction, such as CKD without dialysis, renal clearance of hepcidin is impaired, thus also resulting in the accumulation of hepcidin and contributing to the ongoing disruption of iron absorption and recycling [22].

Low hemoglobin levels (<13 g/dL in men, <12 g/dL in nonpregnant women and children aged >11 years, <11 g/dL in pregnant women and younger children [1]) may indicate insufficient iron available for erythropoiesis (i.e., anemia). In addition, the molecules involved in iron regulation are used as markers of iron availability and iron deficiency. Ferritin is upregulated in the presence of iron [21]; thus, high concentrations may indicate high iron availability, whereas low concentrations (e.g., <15 μg/L in adults) reliably indicate iron deficiency [4,6]. As ferritin is upregulated in inflammatory conditions such as IBD or CKD [21], a higher cut-off may be required (e.g., <30 or <100 μg/L) to define iron deficiency [4,6,24]. Transferrin molecules have two iron-binding sites, but typically the average proportion of overall binding sites occupied (termed transferrin saturation (TSAT)) is 30–50%; a TSAT of ≤20% indicates low iron availability and can be used to diagnose iron deficiency, particularly in the presence of low ferritin levels [4,19,21].

## 2. Iron Replacement Therapy

In patients with iron-deficiency anemia, and in many of those with iron deficiency alone, iron replacement therapy is required in order to support physiologic processes and maintain quality of life, cognitive functioning, and the ability to complete daily activities. It has been estimated that 500 mg of absorbed iron are needed to raise hemoglobin levels by 2 g/dL, which is generally accepted as a meaningful increase in patients with anemia [24]; this amount can be delivered as a single intravenous (IV) iron infusion or as daily oral iron taken for ≥4 weeks [25,26]. However, although this amount of iron addresses the immediate needs of patients with anemia, it may not replenish the body’s iron stores sufficiently to overcome chronic iron deficiency. Particularly in patients with underlying inflammatory diseases, longer-term iron replacement therapy is required to raise and maintain iron stores physiologically, alongside the correction or control of the underlying condition to minimize further loss of iron [5,26].

Oral iron replacement therapy, usually in the form of ferrous salts, offers convenience in terms of low cost and easy administration [27]. However, the effectiveness of many oral formulations may be reduced by limited bioavailability, particularly in patients with underlying inflammatory conditions, because only 10–20% of iron from oral ferrous formulations is estimated to be absorbed [25,28]. Unabsorbed iron can form reactive hydroxyl radicals in the gut [26,29,30,31,32], leading to mucosal irritation or damage [26,29,32,33,34,35,36]. The resulting gastrointestinal adverse events, such as nausea, epigastric discomfort, and constipation, may reduce patients’ willingness to continue treatment [25,26,28,30,35,36]. Unabsorbed iron may also affect the gut microbiome [26,30,32,36,37,38,39] and can trigger disease flares in patients with IBD [26,31,33,36,37]. Absorption may be improved if treatment is taken with ascorbic acid or on an empty stomach to increase or maintain gastrointestinal acidity [26,36]. Tolerability may be improved with concurrent food intake, lower doses, or longer intervals between dosing, although the correction of iron deficiency may be slower as a result [25,26,28,36]. The World Health Organization recommends a maximum oral dosage of elemental iron of 60 mg per day [40], equivalent to approximately 325 mg of ferrous sulfate.

Intravenous iron offers rapid iron replacement by bypassing endogenous iron uptake mechanisms in the gut and is thus useful in patients with significant iron depletion requiring rapid replacement [41]. In terms of immediate iron replacement, IV iron is more effective than oral iron, but there is limited evidence on the long-term impact of IV versus oral iron on healthcare resource use, adverse events, and patients’ quality of life [42]. When using IV iron, physicians must consider the higher cost and resource requirements compared with oral irons. Administration of IV iron may require a hospital or clinic setting, and there are small but potentially serious risks of anaphylaxis, hypophosphatemia, and iron overload (particularly in the presence of functional iron deficiency, when effective processing of available iron is disrupted) [36,43,44].

An easy-to-use oral iron therapy with good absorption and low risk of gastrointestinal adverse events would minimize the burden of treatment on patients requiring long-term iron replacement. Various alternatives to oral ferrous iron formulations have been developed, including polysaccharide–ferric iron complexes, sucrosomial iron, sodium feredate, ferric citrate, and ferric maltol [27,45,46,47].

## 3. Ferric Maltol

Ferric maltol was rationally designed to optimize the absorption and tolerability of oral iron [48]. It has been approved by the European Medicines Agency, SwissMedic, and the US Food and Drug Administration and is commercially available in Europe and the United States for the treatment of adults with iron deficiency, with or without anemia. Regulatory approval was based on a comprehensive program of research, including clinical trials in a variety of settings in adults (IBD, CKD, and pulmonary hypertension). The absorption of iron from ferric maltol has also been investigated in adolescents (aged 10–17 years). To review the evidence base for ferric maltol, we searched PubMed and Google Scholar for articles, excluding patents, published up to 31 May 2021, using the free text search strings (ferric maltol) and (ferric trimaltol), with no date or language restrictions. We supplemented the search results with our own articles in development (now peer-reviewed and published) and congress presentations identified by the manufacturer (Shield Therapeutics (UK) Ltd., Gateshead, UK). We identified 34 publications reporting data on ferric maltol (Appendix A in the Appendix A), including 12 reports of preclinical and clinical pharmacology research [49,50,51,52,53,54,55,56,57,58,59,60], 13 reports of clinical efficacy/effectiveness and safety [61,62,63,64,65,66,67,68,69,70,71,72,73], and 9 reports of health economic or patient-reported outcomes (not reviewed further here) [74,75,76,77,78,79,80,81,82]. While drafting this article, we identified two additional publications, which we have also summarized below [83,84].

### 3.1. Preclinical and Pharmacology Studies

#### 3.1.1. Preclinical Evidence

Ferric maltol is a complex of ferric iron and maltol (3-hydroxy-2-methyl-4-pyrone), a naturally occurring sugar derivative found in many food products, which is highly selective for iron [49]. The iron–maltol complex is stable at a physiologic pH [55] and, as shown in vitro and in vivo, remains strongly chelated until the point of absorption in the gut, when the greater affinity of iron for the iron transport receptor promotes dissociation [50,51,54]. Once dissociated, as demonstrated in vitro, ferric iron is readily transported across the gut lumen [51]. Maltol is separately absorbed, metabolized, and rapidly eliminated in the urine [54].

Studies of ferric maltol in rats have demonstrated that the dissociated iron is absorbed even in models of intestinal damage, with a plateauing effect at higher doses in both healthy and damaged guts, indicating saturable absorption [49,53]. More effective iron absorption from the iron–maltol complex was recorded when iron was in the ferric versus ferrous form [49]. 

Unabsorbed iron remains chelated in the iron–maltol complex until excretion in the feces, thereby potentially reducing the risk of intestinal damage from free iron [53]. Mice with chemically induced colitis had no change in fecal iron content or colitis features after 10 days of treatment with ferric maltol, whereas fecal iron increased significantly, and features of colitis worsened significantly in mice treated with ferrous sulfate [83]. Other researchers have cited the presence of fecal iron as a likely factor in the adverse effects of oral ferrous sulfate therapy in humans [29,85,86], so these preclinical data in mice indicate a potential mechanism for reduced gastrointestinal toxicity with ferric maltol in clinical use. Furthermore, ferric maltol supplementation was associated with fewer changes in the murine microbiome compared with ferrous sulfate. Fecal samples from humans with iron deficiency also showed no significant change in the microbiome from baseline (pretreatment) following treatment with ferric maltol, whereas many genera changed significantly following treatment with ferrous sulfate [83]. These data support a potentially protective effect of ferric maltol on the gut microbiome and an avoidance of free iron-induced intestinal damage.

#### 3.1.2. Clinical Pharmacology

In human studies, iron from ferric maltol was at least as well absorbed as ferrous iron, and it was better absorbed from the complex than from simple ferric salts [55,56]. Iron absorption was increased five-fold when ferric maltol was taken on an empty stomach versus with food [87]. Ferric maltol showed predictable pharmacokinetics with repeated dosing in adults and adolescents (aged 10–17 years) [59,73]. Uptake was rapid, with maximum concentrations (*C*_max_) of iron in the plasma achieved within 2–3 h after administration in adults with IBD [59]. In iron-deficient adults, serum iron levels increased substantially following oral administration of ferric maltol, whereas considerably less absorption was seen in iron-replete adults [56,57], confirming effective absorption of iron with physiologic control of the uptake to meet the body’s needs. Mean ferritin concentrations increased over time, indicating the replenishment of iron stores [59].

In adults with IBD, maltol and maltol glucuronide concentrations increased rapidly and dose proportionally in the plasma (*C*_max_ 1–1.5 h post dose); the ligand was rapidly excreted as maltol glucuronide in the urine within 3 h, and no accumulation was seen with repeated dosing [59]. Similarly, in adolescents (aged 10–17 years), maltol concentrations increased rapidly and dose dependently, and maltol was completely metabolized to maltol glucuronide 2–3 h after administration of the iron–maltol complex [73].

#### 3.1.3. Posology

In Europe and the United States, the approved adult dosage of ferric maltol is 30 mg twice daily (total 60 mg elemental iron/day), to be taken on an empty stomach to maximize absorption. Treatment duration is dependent on the severity of iron deficiency, but generally at least 12 weeks of treatment is required. It is recommended that treatment is continued as long as necessary to replenish the body’s iron stores according to blood tests [88,89,90].

### 3.2. Clinical Evidence Base

#### 3.2.1. Study Designs

Ferric maltol has been studied in phase I–III clinical trials involving 624 adults and 37 adolescents (aged 10–17 years) with iron deficiency [59,61,62,64,68,71,72,73]. Full details of these studies, as well as two reports of real-world experience in 87 adults [69,70], are provided in Appendix A in the Appendix A. Four phase III or IIIB trials in adults (two in patients with quiescent or mild to moderate IBD, one in patients with non-dialysis-dependent stage III or IV CKD, and one in patients with pulmonary hypertension) and a phase I pediatric (adolescent) trial are described here (Table 1) [62,64,68,71,72,73]. In the adult trials, ferric maltol was given at the approved dosage of 30 mg twice daily for ≥12 weeks (≥16 weeks in the CKD study) [62,68,71,72]. In the pediatric study, patients were randomized to receive ferric maltol 7.8, 16.6, or 30 mg twice daily for 10 days [73].

#### 3.2.2. Efficacy Outcomes

The impact of ferric maltol on hemoglobin and iron indices in the phase III trials in adults is summarized in Table 2.

The phase III AEGIS 1/2 IBD and CKD studies met their respective primary endpoints, with statistically significant differences between ferric maltol and the placebo in the change in hemoglobin from baseline to week 12 (*p* < 0.0001) [62] and week 16 (*p* = 0.01) [72], respectively. On average, hemoglobin increased by ≥2 g/dL by week 12 in the AEGIS 1/2 IBD study, which is deemed to be a clinically meaningful rise [62], and the level was sustained up to week 64 [64]. The increase in hemoglobin in the CKD study was smaller, reflecting the complex interaction between inflammation, kidney dysfunction, and iron regulation in these patients [72]. Nevertheless, the increase was significantly greater with ferric maltol than with the placebo (*p* = 0.01), and patients with CKD who were treated for up to 52 weeks achieved ongoing increases in hemoglobin over time [72]. In the phase III IBD head-to-head study, although ferric maltol did not meet the prespecified short-term noninferiority margin versus IV ferric carboxymaltose at the week 12 primary endpoint, the ferric maltol group did achieve a mean increase in hemoglobin of >2 g/dL at that timepoint, which was sustained for up to 52 weeks; at the end of the study, mean hemoglobin levels were similar in the oral and IV arms [71]. These data demonstrate that ferric maltol can provide effective long-term iron replacement and correction of anemia in patients with chronic underlying inflammatory conditions.

In subgroup analyses, ferric maltol had a consistent impact on hemoglobin concentrations regardless of the baseline hemoglobin level, according to a post hoc analysis of the phase IIIB IBD head-to-head study. In the ferric maltol arm, the mean hemoglobin increase from baseline to week 12 was 2.92 g/dL in patients with baseline hemoglobin <9.5 g/dL (baseline mean 8.6 g/dL; *n* = 38) and 2.35 g/dL in patients with baseline hemoglobin ≥9.5 g/dL (baseline mean 10.6 g/dL; *n* = 87), and 70% versus 67%, respectively, achieved a ≥2 g/dL increase or normalization of hemoglobin [84]. The severity of the underlying IBD or CKD activity, as measured by clinical index scores or inflammatory markers such as C-reactive protein [62,65,67], and the use of proton pump inhibitors in patients with IBD [63] did not affect the efficacy of ferric maltol.

Measures of ferritin concentration and TSAT demonstrate that ferric maltol can increase iron availability by week 12 (IBD) and week 16 (CKD) [62,72]. As with hemoglobin, levels of these iron availability markers either increased or were maintained for up to a year with ongoing ferric maltol therapy in patients with underlying IBD or CKD [64,72]. In the IBD head-to-head study, IV ferric carboxymaltose therapy resulted in high ferritin levels at week 12, which had decreased somewhat by week 52; by contrast, levels in the ferric maltol arm increased substantially with ongoing treatment up to week 52, indicating steady replenishment of iron stores over time with this oral therapy [71]. In the phase I pediatric study, iron availability markers were increased at all ferric maltol doses even over the short study duration of 10 days [73].

In the phase IIIB study in patients with pulmonary hypertension (*n* = 22), the mean hemoglobin level increased significantly by 2.9 g/dL, from 10.7 g/dL at baseline to 13.6 g/dL after 12 weeks of treatment with ferric maltol (*p* < 0.001); ferritin and TSAT also increased significantly from baseline to week 12 (both *p* < 0.001) [68]. In the UK real-world FRESH study (*n* = 59), 19 patients achieved normalization of hemoglobin at week 12, and eight further patients achieved normalization after this timepoint; in addition, 16 patients achieved normalization of ferritin [69]. In the single-center, real-world study in London (*n* = 28), mean hemoglobin increased from 11.0 g/dL at baseline to 12.2 g/dL after a median of 16 weeks, while ferritin increased from 14 μg/L at baseline to 28 μg/L after a median of 16 weeks [70].

#### 3.2.3. Safety Findings

In total, 492 adults and adolescents received ferric maltol in clinical studies [59,61,62,64,68,71,72,73], including 345 participants in trials planned to last 12 weeks (IBD and pulmonary hypertension) or 16 weeks (CKD); 293 of these 345 patients (85%) completed 12 or 16 weeks of treatment [62,68,71,72]. In trials with a longer-term follow up, 331 patients received ferric maltol for more than 12 weeks (IBD) or 16 weeks (CKD), of whom 229 (69%) completed treatment for up to 64 weeks (IBD) or 52 weeks (CKD) [64,71,72]. Among patients who stopped treatment before the end of the study, the most common reasons were adverse events (34/96 patients (35%) who stopped ferric maltol prematurely vs. 14/53 patients (13%) who stopped the placebo, and 2/19 patients (11%) who stopped IV ferric carboxymaltose prematurely) and physician or patient decision (28/96 (29%), 18/53 (34%), and 6/19 (32%), respectively) [62,64,68,71,72].

In the phase III studies, the proportion of patients needing to stop ferric maltol therapy because of adverse events before week 12 or 16 was low (~10%) and similar rates in patients randomized to the placebo, even in the AEGIS 1/2 IBD study, which enrolled patients who had been unable to tolerate prior oral ferrous iron therapy [62]. The proportion increased slightly during the longer-term follow up in the AEGIS 1/2 IBD study, but it remained low in the IBD head-to-head and CKD studies [62,64,71,72]. In patients with IBD or CKD, the most common adverse events leading to discontinuation of ferric maltol treatment during long-term treatment (up to 52–64 weeks) were gastrointestinal, including abdominal pain in 2–3% (IBD only), constipation in 1–2%, diarrhea in 1–3%, and nausea in 1–2%. For comparison, among patients with IBD given placebo, 3% stopped prematurely because of abdominal pain and 2% stopped because of diarrhea. Figure 1 summarizes the safety and tolerability of ferric maltol reported in the long-term phase III studies [62,64,68,71,72].

In the exploratory phase IIIB pulmonary hypertension study, treatment with ferric maltol was well tolerated by most patients, with only 2 of 22 patients (9%) unable to complete 12 weeks of treatment (diarrhea *n* = 1, pneumonia *n* = 1) [68]. In the phase I pediatric study, 20 of the 37 adolescents (54%) experienced an adverse event during the 10-day treatment period, with similar frequencies at each ferric maltol dose level. Only one adolescent discontinued treatment because of an adverse event (tonsilitis) [73].

Overall, the most frequent treatment-emergent adverse events were gastrointestinal, which occurred at similar rates in patients treated with ferric maltol or placebo (~30–40%) up to week 12 or 16; the incidence increased slightly with longer treatment (46–57%) in the AEGIS 1/2 IBD and CKD open-label extensions [62,64,71,72]. Gastrointestinal adverse events reported in all studies were constipation (4–6% of patients with IBD, 13–16% of patients with CKD treated with ferric maltol for up to 52–64 weeks, 2–4% of patients with IBD or CKD given placebo for 12–16 weeks, and <1% of patients with IBD treated with ferric carboxymaltose), diarrhea (5–14%, 8–13%, 9–10%, and <1%, respectively), nausea (5%, 12–13%, 2–9%, and 2%, respectively), and vomiting (<1–4%, 0–8%, 3% (IBD only), and 3%, respectively). In addition, nasopharyngitis was reported across all studies (8–18%, 8–11%, 12% (IBD only), and 3%, respectively). Gastrointestinal adverse events reported only in patients with IBD were abdominal pain (9–16% of patients treated with ferric maltol over 52–64 weeks, 12% of patients given placebo for 12 weeks, and 3% of patients treated with ferric carboxymaltose), flatulence (3–8%, 0%, and 0%, respectively), and IBD flare (6–17%, 12%, and 7%, respectively).

In the pulmonary hypertension study, 14% of patients had diarrhea [68]. Gastrointestinal events were also the most common adverse events in the 10-day pediatric study (32%) [73].

Two reports of ferric maltol use in UK clinical practice support the favorable safety profile of ferric maltol in the real world. In the FRESH study, 19 of 59 patients with IBD (32%) experienced adverse events (most commonly abdominal pain/discomfort [15%]), and 30 of 59 patients (51%) were treated for 12 weeks [69]. In the London study, 14 of 21 patients with IBD (67%) tolerated ferric maltol for at least 1 month, including 5 of 10 patients (50%) who had not previously tolerated oral ferrous iron therapy [70].

## 4. Review of the Evidence and Clinical Implications

To date, ferric maltol has been assessed in almost 750 adults and adolescents (661 in clinical trials and 87 in real-world studies) with a range of underlying conditions, including IBD, CKD, and pulmonary hypertension. The clinical trials were designed, as far as possible, to reflect the real-world settings in which ferric maltol will be used, with primary endpoints at 12 or 16 weeks followed by longer-term maintenance treatment in line with recommended treatment durations for oral iron therapies and current understanding of the time required for physiologic restoration of iron stores [45,88,89,90,91]. A further phase III study is being planned in infants and children (age range 1 month to 17 years) with IBD and iron-deficiency anemia, which will use an oral suspension currently under investigation [92].

Underlying inflammatory conditions pose an ongoing risk of iron deficiency; thus, many patients require long-term iron replacement with a well-tolerated iron formulation. Ferric maltol fits this profile, with two-thirds of patients in open-label study arms staying on therapy for up to a year and thus benefitting from the long-term impact of ferric maltol on the body’s iron needs [64,71,72]. Across all of the trial populations studied, ferric maltol has shown consistent and clinically relevant effects on hemoglobin and iron indices, with early increases in hemoglobin to meet erythropoiesis needs, followed by sustained replenishment of iron stores. In patients with IBD, 56–61% achieved a 2 g/dL increase in hemoglobin by week 12. Although ferric maltol has not been directly compared with other oral iron replacement therapies, this is similar to rates reported with oral ferrous sulphate (58–71% at weeks 8–12) [93], and a network meta-analysis indicated favorable hemoglobin improvements with ferric maltol (mean change 2.76 g/dL versus placebo at 12 weeks) compared with other oral irons (mean change 1.04 g/dL) and IV irons (mean change 1.27–2.12 g/dL) after adjusting for baseline hemoglobin levels [66]. Over 52 weeks of treatment, ferric maltol provided similar hemoglobin increases to IV ferric carboxymaltose in patients with IBD, while ferritin levels increased substantially over time [71]. Nevertheless, IV iron should be preferred if a patient has severe anemia or more active IBD (compared with the disease activity in the clinical trials).

In chronic conditions such as iron-deficiency anemia requiring long-term therapy, it is important that treatment does not add to the overall burden of disease. Gastrointestinal adverse events have been widely reported with oral ferrous irons, including the most commonly used of these, ferrous sulphate. In a systematic review and meta-analysis, Tolkien et al. identified 43 trials involving 6831 adults, 3264 of whom received ferrous sulphate (range 7–226 patients/trial). Gastrointestinal adverse events were reported in 2–90% of patients (10–68% in studies where ≥100 patients received ferrous sulphate). The odds ratio for gastrointestinal adverse events in patients who received ferrous sulphate was 2.32 (95% confidence interval 1.74–3.08, *p* < 0.0001) versus placebo and 3.05 (2.07–4.48, *p* < 0.001) versus intravenous iron [30]. In the AEGIS 1/2 IBD study, the proportion of patients experiencing any adverse events up to week 12 was lower in the ferric maltol arm (58%) than in the placebo arm (72%), while the proportion of patients with gastrointestinal events was similar (38% and 40%, respectively) [62]. In the CKD study, the overall incidence of adverse events was again lower with ferric maltol (68%) than with the placebo (75%) up to week 16, although the incidence of gastrointestinal events was higher (40% vs. 31%) [72]. The incidence of any adverse events and of gastrointestinal events increased slightly with longer-term treatment (up to 52 or 64 weeks) in both studies [64,72]. In the IBD head-to-head study versus IV ferric carboxymaltose, 10% of patients in the ferric maltol arm stopped treatment because of adverse events compared with 3% in the IV arm. The frequency of gastrointestinal events was higher with ferric maltol (31% up to week 52) than with IV iron (13%), but the incidence in the ferric maltol arm was consistent with that reported with the placebo in the AEGIS 1/2 IBD study [62,71].

In a meta-analysis of IV versus oral ferrous irons, Bonovas et al. reported an odds ratio of 0.24 (0.12–0.49) in favor of IV iron for treatment discontinuation due to adverse events, even though most of the oral iron studies may have had a selection bias toward favorable tolerability by excluding patients with known intolerance to previous oral irons [94], highlighting the need for a better tolerated oral iron formulation. In the AEGIS 1/2 IBD study, two-thirds of patients had previously stopped oral ferrous iron therapy because of adverse events; thus, they might be expected to have poor tolerance of a subsequent oral iron therapy, but in fact most patients were willing to continue ferric maltol for up to 64 weeks [62,64]. This finding is supported by the proof-of-concept study, in which 19 of 23 patients with known intolerance of prior oral therapies completed 3 months of ferric maltol therapy [61], and the reported real-world experience, in which 50% of patients, including a high proportion with prior oral therapy intolerance, were able to tolerate ferric maltol [69,70].

Collectively, these findings indicate that prolonged treatment with ferric maltol provides sufficient iron to meet the body’s erythropoietic needs, which could help reduce the need for erythropoiesis-stimulating agents or blood transfusions. At the same time, the amount of free iron in the gut is minimized, thereby reducing the risk of damage to the gut microbiome and of exacerbation of any underlying gastrointestinal disease.

## 5. Summary and Future Directions

The oral iron replacement therapy ferric maltol is licensed in Europe and the United States for the treatment of adults with iron deficiency with or without anemia. Of the 661 adults and adolescents who participated in the clinical trials, 492 received ferric maltol, and it has been widely used in clinical practice. Consistent evidence in a range of settings, including IBD, CKD, and pulmonary hypertension, indicates that ferric maltol is an effective and efficacious oral therapy for patients with iron deficiency and anemia, regardless of the underlying disease. The clinical impact of ferric maltol on iron deficiency without anemia or in other settings should be confirmed in future research. Future head-to-head studies of different oral iron replacement therapies in disease conditions where a direct comparison might be appropriate and ethically feasible especially in the context of drug safety could also help to guide clinical decision making. Nevertheless, with the exception of patients experiencing an IBD flare [88,89,90] and those requiring rapid iron replacement (best achieved with IV iron), we believe that ferric maltol is an appropriate treatment option for patients in whom long-term, convenient, and well-tolerated management of chronic iron deficiency is desired.

## Figures and Tables

**Figure 1 jcm-10-04448-f001:**
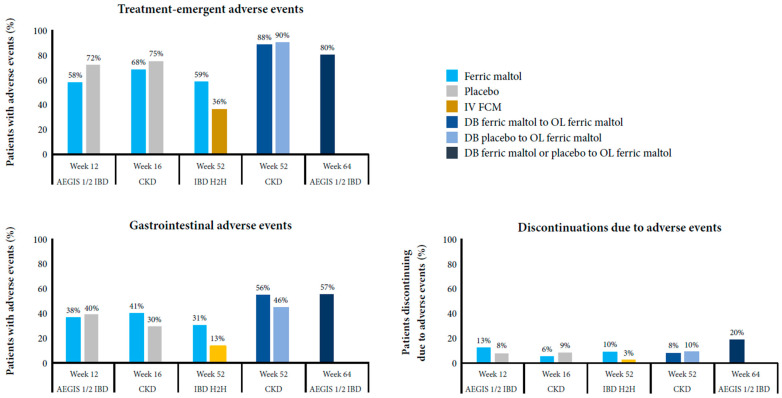
Safety and tolerability of ferric maltol in the long-term phase III studies [62,64,71,72].

**Table 1 jcm-10-04448-t001:** Designs of key ferric maltol clinical studies.

Study	Location	Underlying Condition	Anemia and Iron Deficiency Definitions	Patients, N	Design	Comparator	Primary Endpoint
Adult							
Phase III IBD (AEGIS 1/2) [62,64]	Global	Eligibility criteria: Quiescent or mild or moderate IBD UC: SCCAI score <4 at screening and randomization CD: CDAI score <220 at randomization At baseline: UC: FM *n* = 29; placebo *n* = 29 Median (range) SCCAI score: FM 2.0 (0–3); placebo 2.0 (0–3) CD: FM *n* = 35; placebo *n* = 35 Median (range) CDAI score: FM 75 (14–199); placebo 108 (10–220)	Hb ≥9.5 to <12.0 g/dL (women) or <13.0 g/dL (men) Ferritin <30 μg/L at screening	128 (FM *n* = 64; placebo *n* = 64) 97 started OL FM after DB FM (*n* = 50) or DB placebo (*n* = 47)	Randomized, DB, superiority52-week OL extension	Placebo (DB period only)	Hb change from baseline to week 12
Phase IIIB IBD (H2H) [71]	Global	Eligibility criteria: Quiescent or mild or moderate IBD UC: SCCAI score ≤5 during screening CD: CDAI score ≤300 during screening At baseline: UC: FM *n* = 46, IV FCM *n* = 46 Mean (SD) SCCAI score: FM 2.2 (1.8); IV FCM 2.3 (1.6) CD: FM *n* = 79, IV FCM *n* = 79 Mean (SD) CDAI score: FM 129.6 (60.1); IV FCM 140.5 (75.8)	Hb ≥8.0 to ≤11.0 g/dL (women) or ≤12.0 g/dL (men) Ferritin <30 μg/L or ferritin <100 μg/L + TSAT <20%	250 (ITT: FM *n* = 125, IV FCM *n* = 125; PP: FM *n* = 78, IV FCM *n* = 88)	Randomized, OL, non-inferiority	IV FCM	Hb responder rate at week 12 (≥2 g/dL increase or normalization)
Phase III CKD [72]	USA	Eligibility criteria: CKD stage III or IV (eGFR ≥15 to <60 mL/min/1.73 m^2^, not on dialysis) At baseline:Mean (SD) eGFR: FM 31.9 (11.5) mL/min/1.73 m^2^ Placebo 29.7 (10.6) mL/min/1.73 m^2^	Hb ≥8.0 to <11.0 g/dL Ferritin <250 μg/L + TSAT <25% or ferritin <500 μg/L + TSAT <15%	167 (FM *n* = 111; placebo *n* = 56) 125 started OL FM after 16 weeks of DB FM (*n* = 86) or DB placebo (*n* = 39)	Randomized, DB, superiority36-week OL extension	Placebo	Hb change from baseline to week 16
Phase IIIB PH [68]‘	Germany	Eligibility criteria: any form of PH with mean resting pulmonary artery pressure ≥25 mmHg At baseline: PAH (*n* = 14) PH due to left heart disease (*n* = 1) Inoperable chronic thromboembolic PH (*n* = 7) Mean (SD) pulmonary artery pressure 50 (11) mmHg	Hb ≥7 to <12 g/dL (women) or ≥8 to <13 g/dL (men) Ferritin <100 μg/L or ferritin 100–300 μg/L + TSAT <20%	22	Single-arm OL, exploratory	None	Hb change from baseline to week 12
Pediatric							
Phase I [73]	UK	Eligibility criteria: Iron deficiency of any cause At baseline: CD (*n* = 8) Other gastrointestinal disorders (*n* = 11) Vitamin D deficiency (*n* = 7) CKD (*n* = 4) Other conditions (*n* = 7)	Ferritin <30 μg/L or ferritin <50 μg/L + TSAT <20%	37	Randomized, exploratory	Different FM doses	PK, iron uptake

CD, Crohn’s disease; CDAI, Crohn’s Disease Activity Index; CKD, chronic kidney disease; DB, double-blind; eGFR, estimated glomerular filtration rate; FCM, ferric carboxymaltose; FM, ferric maltol; H2H, head-to-head; Hb, hemoglobin; IBD, inflammatory bowel disease; ITT, intention-to-treat; IV, intravenous; OL, open-label; PAH, pulmonary arterial hypertension; PH, pulmonary hypertension; PK, pharmacokinetics; PP, per protocol; SCCAI, Simple Clinical Colitis Activity Index; SD, standard deviation; TSAT, transferrin saturation; UC, ulcerative colitis.

**Table 2 jcm-10-04448-t002:** Phase III studies of ferric maltol in adults: key efficacy outcomes.

Study	Mean Hb and Change from Baseline	Hb Responder Rate ^1^	Proportion of Patients Achieving Hb Normalization ^2^	Proportion of Patients Achieving ≥2 g/dL Increase in Hb	Mean Ferritin	Mean TSAT
Phase III IBD (AEGIS 1/2) [62,64]	Baseline FM: 11.0 g/dL Placebo: 11.1 g/dL Week 12 FM: 13.2 g/dL Placebo: 11.2 g/dL Mean (SE) difference in change from baseline to week 12 FM vs. placebo: 2.25 (0.12) g/dL *p* < 0.0001 Up to week 64 DB FM to OL FM: 13.95 g/dL DB placebo to OL FM: 13.33 g/dL	NR	Week 12 FM: 66% Placebo: 13% Up to week 64 DB FM/placebo to OL FM: 86%	Week 12 FM: 56% Placebo: 0 Up to week 64%NR	Baseline FM: 8.6 μg/L Placebo: 8.2 μg/L Week 12 FM: 26.0 μg/L Placebo: 9.8 μg/L Mean increase at week 12 FM: 17.3 μg/L Placebo: 1.2 μg/L Up to week 64 DB FM/placebo to OL FM: 57.4 μg/L	Baseline FM: 10.6% Placebo: 9.5% Week 12 FM: 28.5% Placebo: 9.8% Mean increase at week 12 FM: 18.0 percentage points Placebo: −0.4 percentage points Up to week 64 DB FM/placebo to OL FM: 29%
Phase IIIB IBD (H2H) [71]	ITT population Baseline FM: 10.0 g/dL IV FCM: 10.1 g/dL Week 12 FM: 12.5 g/dL IV FCM: 13.2 g/dL LSM difference (95% CI) between groups at week 12 FM–IV FCM: −0.6 (−1.0 to −0.2) g/dL *p* = 0.002 Up to week 52/EoT FM: 12.8 g/dL IV FCM: 13.0 g/dL	ITT population Week 12 FM: 67% IV FCM: 84% Risk difference (95% CI) FM–IV FCM: −0.17 (−0.28 to −0.06) PP population Week 12 FM: 68% IV FCM: 85% Risk difference (95% CI) FM–IV FCM: −0.17 (−0.30 to 0.05) ^3^	ITT population Week 12 FM: 55% IV FCM: 81% Up to week 52/EoT NR	ITT population Week 12 FM: 61% IV FCM: 77% Up to week 52/EoT NR	ITT population Baseline FM: 16.6 μg/L IV FCM: 9.2 μg/L Week 12 FM: 25.7 μg/L IV FCM: 139.2 μg/L LSM difference (95% CI) between groups at week 12 FM–IV FCM: −113.1 (−145.9 to –80.2) μg/L *p* < 0.001 Up to week 52/EoT FM: 78.9 μg/L IV FCM: 103.4 μg/L	NR
Phase III CKD [72]	Baseline FM: 10.1 g/dL Placebo: 10.0 g/dL LSM change from baseline to week 16 FM: 0.5 g/dL Placebo: −0.0 g/dL LSM (SE) difference between groups at week 16 FM–placebo: 0.5 (0.2) g/dL *p* = 0.01 Up to week 52/EoT DB FM to OL FM: 10.9 g/dL DB placebo to OL FM: 10.9 g/dL	NR	Week 16 FM: 27% Placebo: 13% Up to week 52/EoT NR	Week 16 FM: 6% Placebo: 0 Up to week 52/EoT NR	Baseline FM: 97.0 μg/L Placebo: 104.2 μg/L LSM change from baseline to week 16 FM: 25.4 μg/L Placebo: −7.2 μg/L LSM (SE) difference between groups at week 16 FM–placebo: 32.7 (9.4) μg/L *p* < 0.001 Up to week 52/EoT DB FM to OL FM: 142.5 μg/L DB placebo to OL FM: 146.3 μg/L	Baseline FM: 15.7% Placebo: 15.6% LSM change from baseline to week 16 FM: 3.8 percentage points Placebo: −0.9 percentage points LSM (SE) difference between groups at week 16 FM–placebo: 4.6 (1.1) percentage points *p* < 0.001 Up to week 52/EoT DB FM to OL FM: 23.5% DB placebo to OL FM: 21.4%
Phase IIIB PH [68]	Baseline FM: 10.7 g/dL Week 12 FM: 13.6 g/dL Median increase from baseline to week 12: 2.9 g/dL *p* < 0.001	NR	NR	NR	Baseline FM: 13.1 μg/L Week 12 FM: 36.6 μg/L *p* < 0.001	Baseline FM: 7.5% Week 12 FM: 31.7% *p* < 0.001

^1^ Responder rate was defined as an increase in Hb of ≥2 g/dL from baseline and/or normalization of Hb (≥12 g/dL in women, ≥13 g/dL in men) [71]. ^2^ Hb normalization was defined as an increase in Hb to ≥12 g/dL in women or ≥13 g/dL in men in the IBD trials or to >11 g/dL in the CKD trial [62,71,72]. ^3^ Hb responder rate at week 12 was the primary endpoint for the IBD H2H study; as the CIs crossed the prespecified noninferiority margin in the ITT and PP analyses, the primary endpoint of noninferiority of FM vs. IV FCM was not met [71]. CI, confidence interval; EoT, end of treatment; LSM, least-squares mean; NR, not reported; SE, standard error.

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
