# Peer review of "Iron Replacement Therapy with Oral Ferric Maltol: Review of the Evidence and Expert Opinion"

_jcm, 2021, doi:10.3390/jcm10194448_

Round 1

Reviewer 1 Report

The authors have performed an extensive review dealing with the mode of action of ferric maltose. They have described all major studies comparing this formulation to placebo. It seems that direct comparisons to other formulations are lacking. Hower, ferric-maltose appears to be effective for iron deficiency anemia. In my view data on CKD and pulmonary hypertension are quite redundant as the IBD patient population is unique and I would focus on that.

Specific comments:

In my view the fact that Shield Therapeutics (UK) Ltd (Gateshead, UK) paid for medical writing assistance should also be mentioned at the abstract or a as a clearer conflict of interest.

I would shorten the introdution - focus more on etiology, prevalence and cosequences of iron deficiency in IBD patients, and less on iron deficiency in general and on the specific pathways and carrier proteins involved.

Page 4, lines 177-180 - Is there evidence of the effect of ferrous-sulphate on fecl iron secretion? In my view this is worthwhile mentioning data from human studies, especially that ferrous-sulphate is extensively used worldwide.

Please mention efficacy rates with other oral iron supplements.

3.2.3. Safety findings

Please elaborate the adverse events reported with ferric-maltose. Please compare to rate of adverse events reported with ferrous-sulphate.

4. Clinical recommendations - I agree that ferric-maltose seems a feasible option for oral iron supplementation. however, corroborating studies ought to be performed in order to determine if this formulation is superior to other supplements.

Reviewer 2 Report

The authors present an excellent review focusing on iron replacement with Ferric maltol summarizing available studies till May 2021. 

The introduction includes a general overview of iron deficiency and a section on iron replacement therapy. There is a summary of Ferric maltol and preclinical studies.  

The authors included key clinical studies and presented both design and efficacy outcomes in tables. There was also a description of safety and tolerability findings in longer-term studies. 

I enjoyed the review, which I found well written, good distribution, and easy to follow. 

I have a couple of observations.

-In the section on iron replacement therapy, the authors comment on oral iron, focusing on ferrous salts. The authors could at least mention other types of oral iron.

-In the Safety findings, the authors comment on the overall GI events. It would be interesting to mention which GI events were the most common cause of treatment disruption, mainly since some of the clinical studies include IBD patients; for example, epigastric pain might have a different impact than diarrhea in these patients.   
